# *Arabidopsis thaliana* MYC2 and MYC3 Are Involved in Ethylene-Regulated Hypocotyl Growth as Negative Regulators

**DOI:** 10.3390/ijms25158022

**Published:** 2024-07-23

**Authors:** Yuke Li, Ying Cheng, Fan Wei, Yingxiao Liu, Ruojia Zhu, Pingxia Zhao, Jing Zhang, Chengbin Xiang, Erfang Kang, Zhonglin Shang

**Affiliations:** 1Ministry of Education Key Laboratory of Molecular and Cellular Biology, Hebei Collaboration Innovation Center for Cell Signaling and Environmental Adaptation, Hebei Research Center of the Basic Discipline of Cell Biology, Hebei Key Laboratory of Molecular and Cellular Biology, College of Life Sciences, Hebei Normal University, Shijiazhuang 050024, China; kk20231219@163.com (Y.L.); chengyingdpl@163.com (Y.C.); 15525119212@163.com (F.W.); 15703277922@163.com (Y.L.); 2College of Pharmacy, Hebei University of Chinese Medicine, Shijiazhuang 050200, China; zhuruojia1990@126.com; 3Division of Life Sciences and Medicine, Division of Molecular & Cell Biophysics, Hefei National Science Center for Interdisciplinary Sciences at the Microscale, MOE Key Laboratory for Membraneless Organelles and Cellular Dynamics, University of Science and Technology of China, Hefei 230026, China; zpx168@ustc.edu.cn (P.Z.); zj8155@ustc.edu.cn (J.Z.); xiangcb@ustc.edu.cn (C.X.)

**Keywords:** *Arabidopsis thaliana*, MYC2, MYC3, ethylene, hypocotyl, ERF1

## Abstract

The ethylene-regulated hypocotyl elongation of *Arabidopsis thaliana* involves many transcription factors. The specific role of MYC transcription factors in ethylene signal transduction is not completely understood. The results here revealed that two MYCs, MYC2 and MYC3, act as negative regulators in ethylene-suppressed hypocotyl elongation. Etiolated seedlings of the loss-of-function mutant of MYC2 or MYC3 were significantly longer than wild-type seedlings. Single- or double-null mutants of MYC2 and MYC3 displayed remarkably enhanced response to ACC(1-aminocyclopropane-1-carboxylate), the ethylene precursor, compared to wild-type seedlings. MYC2 and MYC3 directly bind to the promoter zone of ERF1, strongly suppressing its expression. Additionally, EIN3, a key component in ethylene signaling, interacts with MYC2 or MYC3 and significantly suppresses their binding to ERF1’s promoter. MYC2 and MYC3 play crucial roles in the ethylene-regulated expression of functional genes. The results revealed the novel role and functional mechanism of these transcription factors in ethylene signal transduction. The findings provide valuable information for deepening our understanding of their role in regulating plant growth and responding to stress.

## 1. Introduction

During seed germination, the elongation of the hypocotyl pushes the primary root out of the seed coat and allows it to absorb water and minerals from the soil. Hypocotyl elongation also helps in uncovering the cotyledons, which is essential for photomorphogenesis and the development of autotrophic seedlings [1,2]. The hypocotyl is sensitive to various internal and external signals [3], making it an ideal model for studying the regulation of plant growth and development [4,5].

When elongating hypocotyls encounter obstacles, ethylene levels increase to suppress the elongation rate and boost the jacking force, which is necessary for pushing seedlings out of the soil [6]. In ethylene-regulated growth and development, five ethylene receptors in the endoplasmic reticulum (ER) membrane, including ETR1, ERS1, ETR2, ERS2, and EIN4, are involved in responding to ethylene and triggering downstream signaling events [7,8,9]. When ethylene is absent, these receptors bind to CTR1 to form a complex. As a serine/threonine protein kinase, CTR1 phosphorylates EIN2. Since phosphorylated EIN2 is unable to trigger EIN3/EIL1, which act as transcription factor, the signaling cascade is blocked. When the ethylene content increases, it binds and disassembles the receptor complex, releasing the inhibitory effect of CTR1 on EIN2. Activated EIN2 migrates from the ER membrane to the nucleus and stimulates EIN3/EIL1, which ultimately alters functional gene expression related to growth and development [9,10,11].

The growth of the Arabidopsis hypocotyl is influenced by ethylene in relation to light exposure. Ethylene suppresses elongation of etiolated hypocotyls in the dark but promotes elongation in the light [3,12]. In darkness, hypocotyls of the ethylene-overproducing mutant *eto1-1* and the null mutant of *CTR1* (*ctr1-1*) are shorter than that of the wild type, whereas hypocotyls of the ethylene-insensitive mutants *ein2-5* and *ein3eil1* are longer than that of the wild type [13,14]. In the light, hypocotyls of the *EIN3*-overexpression line and *ctr1-1* are longer than that of the wild type, whereas hypocotyls of *ein2-5* and *ein3eil1* are shorter than that of the wild type [12]. The expression of Ethylene Response Factor 1 (ERF1) is regulated by ethylene, and overexpression of ERF1 significantly inhibits elongation of etiolated hypocotyls, thereby positively regulating ethylene-inhibited elongation [15,16].

MYC transcription factors belong to the plant bHLH transcription factor family. The first plant MYC gene was discovered in maize, and it is involved in regulating anthocyanin biosynthesis [17]. In Arabidopsis, AtMYC2 and its three homologues, AtMYC3, AtMYC4, and AtMYC5, have been extensively studied [18,19]. MYCs are involved in four physiological processes: (1) The first is plant growth and development, in which MYC2, MYC3, MYC4, and MYC5 inhibit cell proliferation of integument, thus negatively regulating seed enlargement [20]. MYC3 interacts with the DELLA protein, antagonizes the regulation of CO (CONSTANS) on *FT* (FLOWERING LOCUS T) gene expression, and participates in the flowering process, which is regulated by photoperiod and gibberellin (GA) [21]. MYC2, MYC3, and MYC4 negatively regulate jasmonic acid (JA)-inhibited root and hypocotyl growth by regulating the expression of *HY5* [22,23]. MYC2 promotes the expression of a senescence-related gene, *Dof2.1*, thus enhancing JA-induced leaf senescence [24]. (2) The second is plants’ response to biotic stresses. In *Solanum lycopersicum*, magnesium oxide triggers the expression of resistance genes by stimulating JA signaling and activating MYC2, thus inducing the immune response to *Fusarium* wilt [25]. In rice, overexpression of *OsMYC2* induces up-regulation of pathogen resistance genes and enhances resistance to bacterial blight [26]. In Arabidopsis, AtPIFs and AtMYC2 form a homodimer that binds the promoter of the *TPS* (terpene synthase) gene. An increase in terpene biosynthesis enhances the defensive ability of whitefly (*Bemisia tabaci*) [27]. (3) The third is plants’ response to abiotic stresses. MYC1 inhibits the binding of FIT (FER-like iron deficiency-induced transcription factor) with bHLH38/39 heterodimer, suppressing the expression of its target genes (*FRO2* and *IRT1*), thus inhibiting iron absorption and tolerance of iron deficiency [28]. MYC2 enhances Arabidopsis’ tolerance to salinity by regulating proline biosynthesis [29]. The MYC2/MYC3/MYC4-dependent transcription factor network regulates water spray-responsive gene expression as well as JA accumulation, responding to water spray stress caused by simulated rain [30]. (4) The last process involves regulating the synthesis of secondary metabolites. Some MYCs regulate gene expression of components in the biosynthesis of tryptophan, alkaloids, or flavonoids, promoting the accumulation of these secondary metabolites [26,31,32]. The accumulation of flavonoids inhibits the growth of insects and enhances the recovery ability of plant cells, increasing resistance to insect attacks [33].

The growth and development of plants are regulated by phytohormones, which also induce stress response by stimulating or suppressing the expression of functional genes using multiple transcription factors. In ethylene signaling, several agents can bind and regulate the activity of some bHLH transcription factor members. Phytochrome-interacting factors (PIFs), a group of well-characterized bHLH transcription factors, are involved in the cross-talk of ethylene- and photosignaling [34]. Overexpression of *PIF5* increases ethylene levels in etiolated seedlings by stimulating the expression of ACC synthase [35,36]. Ethylene promotes the hypocotyl elongation of green seedlings by increasing *PIF3* expression [12]. PIF1 and EIN3/EIL1 inhibit photobleaching by inhibiting the expression of protochlorophyllide biosynthetic genes and activating the expression of *POR* genes [37]. EIN3 and EIL1 participate in skotomorphogenesis by regulating the activity of PIF3 [16]. PIFs and EIN3/EIL1 are involved in apical hooks development, which is regulated by phytohormones (e.g., ethylene, gibberellin, JA, salicylic acid, and cytokinin) and light, by regulating the expression of *HLS140* [38,39,40]. MYCs are involved in the antagonism of JA- and ethylene signaling in *Arabidopsis*. MYC2/3/4 repress the transcriptional activity of EIN3/EIL1, thus suppressing ethylene-promoted resistance to necrotic fungi and apical hook formation [41]; vice versa, EIN3/EIL1 inhibits the transcriptional activity of MYC2/3/4 to inhibit the expression of wound response genes (*TAT3*, *VSP1*, and *VSP2*) and herbivore-induced genes (*CYP79B3*, *BCAT4*, and *BAT5*) induced by JA [42].

Several MYCs have been studied for their physiological functions and functional mechanisms, but the specific role of MYC2 and MYC3 in ethylene signaling is not fully understood. This study confirms the involvement of MYC2 and MYC3 in regulating hypocotyl growth in response to ethylene and explores their relationship with various ethylene signaling agents. The findings offer new insights into the role of MYC2 and MYC3 in the ethylene signal transduction pathway and shed light on the mechanism behind ethylene-regulated seedling growth.

## 2. Results

### 2.1. MYC2 and MYC3 Negatively Regulate the Ethylene-Inhibited Elongation of Etiolated Hypocotyls

To analyze the role of MYC3 and MYC3 in ethylene-inhibited etiolated hypocotyl elongation, the response of seedlings, including Col-0 and two MYC2 single-null mutants (*myc2-1* and *myc2-2*) and an MYC3-null mutant (*myc3*), to ethylene precursor ACC was analyzed. The hypocotyl length of *myc2-1*, *myc2-2*, and *myc3* seedlings were 13.86 ± 0.74, 14.00 ± 1.07, and 13.84 ± 1.09 mm, respectively, significantly longer than the hypocotyl of Col-0 seedlings (11.83 ± 0.74 mm) (*p* < 0.05) (Figure 1A), indicating that MYC2 and MYC3 may play negative roles in the elongation of etiolated hypocotyls. The etiolated hypocotyl of the double-null mutant (*myc2myc3*) seedlings was longer than that of the *myc2* and *myc3* seedlings (*p* < 0.01), indicating that MYC2 and MYC3 may regulate hypocotyl elongation synergistically (Figure 1A,B).

On a medium containing ACC, the growth rate of etiolated hypocotyls significantly decreased. The effect of 25 μΜ ACC was stronger than that of 10 μΜ ACC, and the effect of 50 μΜ ACC was similar to that of 25 μΜ ACC. The length of the etiolated hypocotyls of *myc2-1*, *myc2-2*, *myc3*, and *myc2myc3* seedlings was similar to that of Col-0 seedlings after treatment with 10, 25, or 50 μΜ ACC (*p* > 0.05) (Figure 1B). To clarify the role of MYC2 or MYC3 in ethylene-inhibited etiolated hypocotyl elongation, the ratio of etiolated hypocotyl length under ACC to the control was calculated. The results showed that the ratio of any null mutant was higher than that of Col-0, indicating that mutants respond to ACC more sensitively than the wild type (Figure 1C).

To elucidate the role of MYCs in etiolated seedlings’ responses to ethylene, the growth parameters of seedlings that were cultured on ACC-containing medium were measured. Notably, 0.1 μΜ ACC significantly inhibited the elongation of the etiolated hypocotyls of *myc2-2*, *myc3*, and *myc2myc3*, while it did not affect etiolated seedlings of *myc2-1* and Col-0. On 0.25 μΜ ACC-containing medium, all *myc* mutants’ seedlings were notably suppressed, while Col-0 seedlings were only slightly suppressed. Furthermore, on a medium containing 0.5 μΜ ACC, the elongation of hypocotyls in *myc2myc3* displayed increased sensitivity compared to Col-0 and the three single-null mutants. Col-0 and all mutants’ seedlings responded to 0.75 or 1 μΜ ACC in a similar manner and to a similar extent. The results indicate that the effectiveness of MYC2 and MYC3 concerning the etiolated hypocotyls’ response to ethylene depends on ethylene dosage (Figure 1D,E).

To further verify the role of MYC2 and MYC3 in ethylene-regulated seedling growth, the response of the etiolated seedlings of MYC2 or MYC3 complementation lines to ACC was detected. The results showed that seedlings of two MYC2 complementation lines (*MYC2#COM1* and *MYC2#COM2*) and two MYC3 complementation lines (*MYC3#COM1* and *MYC3#COM2*) responded to ACC with similar sensitivity as the Col-0 seedlings (Appendix A).

It was reported that while ethylene suppresses the elongation of etiolated hypocotyls, it promotes the elongation of hypocotyls in green seedlings [3,12]. To further investigate the role of MYC2 and MYC3 in ethylene-regulated seedling growth, seedlings were cultured under light on ACC-containing medium. The results showed that the seedling growth rate was similar in Col-0 and *MYC2-* or MYC3-null mutants. In other words, the hypocotyls of Col-0, *myc2-1*, *myc2-2*, *myc3*, and *myc2myc3* seedlings were all significantly promoted by ACC (Appendix A), suggesting that MYC2 and MYC3 may not be involved in the ethylene-promoted elongation of hypocotyls in green seedlings.

To further investigate the role of MYC2 and MYC3 in ethylene-suppressed etiolated hypocotyl elongation, scanning electron microscopy (SEM) was used to detect etiolated hypocotyl cells. The epidermal cell count showed that the number of hypocotyl cells in both Col-0 and all mutant seedlings was similar, with each row containing 20–22 epidermal cells. Etiolated seedlings grown on ACC-free medium showed that the hypocotyl cell length of *myc2-1*, *myc2-2*, and *myc3* were significantly longer than that of Col-0 but shorter than that of *myc2myc3*. On ACC-containing medium, the hypocotyl cell length decreased significantly, and the cell length of *myc2-1*, *myc2-2*, and *myc3* was not significantly different from that of Col-0 (Figure 2). This suggests that mutants were more sensitive to ACC than the wild type, indicating that MYC2 and MYC3 may play a negative role in ethylene-regulated cell elongation in etiolated hypocotyls.

### 2.2. MYC2 and MYC3 Inhibit Expression of ERF1 by Binding Its Promoter

To verify the possible mechanism of MYC2 and MYC3-regulated hypocotyl growth, the expression of ERF1 in *myc2* or *myc3* was analyzed. In 4-days-old etiolated seedlings, the expression of ERF1 in *myc2*, *myc3* was markedly higher than in Col-0 (*p* < 0.01), and the expression level in *myc2-2* was the highest among the three mutants, which was 7–8 times that of Col-0 (Figure 3A). The results indicate that ERF1 expression is negatively regulated by MYC2 or MYC3.

MYC2 and MYC3 can bind G-box, a cis-acting element in the promoter of target genes, to regulate gene expression [23]. In the promoter of ERF1, there are four potential MYC2-binding sites located at −1233~−1238 (AACATG), −1279~−1284 (CACATG), −1830~−1835 (CATGTG), and −1921~−1926 (CACGTG) upstream of the transcription start point and two potential MYC3-binding sites located at −1789~−1794 (ATCCGT) and −1921~−1926 (CACGTG) upstream of transcription start point, respectively (Figure 3B). MYC2 or MYC3 may bind these elements and regulate ERF1 expression.

To verify the binding of MYC2 or MYC3 with the ERF1 promoter, GST-MYC2 and GST-MYC3 proteins were purified. Probe primers were designed based on the potential binding sequence in the ERF1 promoter, and EMSA (electrophoretic mobility shift assay) was performed. The results showed that GST-MYC2 or GST-MYC3 bound to P1 (Figure 3C,D), while they did not bind to P2, P3, P4, or P5 of the ERF1 promoter (Appendix A). The binding was abolished by adding an unlabeled P1 probe, but it was not affected by mutated probes.

To confirm the binding of MYC2 or MYC3 with the ERF1 promoter, MYC2 or MYC3 was cloned to *pSuper::pCAMBIA1300GFP* binary vector and transformed into Col-0, and DNA was extracted and purified from transformant seedlings for chromatin immunoprecipitation (ChIP) assay, which was performed by using GFP (green fluorescent protein) antibody. The results showed that ERF1 promoter sequences in genomic DNA fragments were co-immunoprecipitated by MYC2 or MYC3 (Figure 3E,F), proving the in vivo binding of MYC2 or MYC3 with ERF1 promoter.

To evaluate the effect of MYC2 or MYC3 on the expression of ERF1, the ERF1 promoter-driven GUS (β-glucuronidase) reporter was transiently expressed. *pSuper::MYC2* and *pSuper::MYC3* were used as effectors. GUS vectors driven by ERF1 promoter, including G-box or mutated G-box, were used as reporters (Figure 4A). Genes were transiently expressed in *Nicotiana benthamiana* leaves. The results showed that MYC2 or MYC3 inhibited ERF1 expression through the G-box region in the ERF1 promoter since the inhibitory effect was not detected when G-box was mutated (Figure 4B,C).

### 2.3. EIN3 Suppresses the Binding of MYC2 or MYC3 with ERF1’s Promoter

Interaction between some MYCs (e.g., MYC2, MYC3, and MYC4) and EIN3 has been reported [41,42]. The result of yeast two-hybridprimarily showed that MYC2 and MYC3 did not interact with EIN2 (Appendix A). It was reported that EIN3 can bind ERF1’s promoter [15]. To investigate the impact of EIN3 on the transcriptional activities of MYC2 or MYC3, fusion genes including *EIN3-MYC*, *MYC2-GFP*, *MYC3-GFP*, *ERF1_pro_::GUS*, and *ERF1_mpro_::GUS* were combined and co-expressed separately in *N. benthamiana* leaves (Figure 4A). The results indicated that the expression of ERF1 decreased when MYC2 or MYC3 were transiently expressed in *N. benthamiana* leaves. Moreover, when the EIN3 binding sites were mutated and co-expressed with MYC2 or MYC3, the inhibitory effect of MYC2 or MYC3 on ERF1 expression was further strengthened. When P1 was mutated and co-expressed with *EIN3*, the GUS activity increased, indicating that EIN3 acts on the ERF1 promoter independently of MYCs; i.e., MYCs and EIN3 regulate the expression of ERF1 through different sites. Additionally, when both the EIN3 binding sites and P1 were mutated, the expression of ERF1 remained unaltered (Figure 4B,C), suggesting that EIN3 hinders the transcriptional activity of MYC2 or MYC3. The results from EMSA showed that EIN3 inhibits the binding of MYC2 or MYC3 to ERF1’s promoter. Furthermore, with an increasing dosage of EIN3, the binding of MYC2 to P1 was gradually weakened (Figure 4D,E).

### 2.4. MYC2 and MYC3 Genetically Act Upstream of ERF1

To verify the role of ERF1 in ethylene-suppressed etiolated hypocotyl elongation, the response to ACC of four ERF1-null mutant seedlings (*erf1-1* and *erf1-2* were generated using CRISPR/Cas9, as shown in Figure 5A; *erf1-3* and *erf1-4* were generated using RNAi) and two ERF1-overexpression lines (*ERF1-OE1* and *ERF1-OE2*) was detected. On ACC-free medium, the hypocotyl length of the ERF1-null mutants was similar to that of the wild type, while the hypocotyls of the *ERF1-OE* lines were significantly shorter than those of Col-0. On ACC-containing medium, the hypocotyl length of the null mutants’ seedlings was similar to that of Col-0, while the hypocotyls of the *ERF1-OE* lines were significantly shorter than those of Col-0 (Figure 5B,C). The ratio of hypocotyl length for seedlings grown on ACC/ACC-free medium showed that the ratios of *ERF1-OE* lines were significantly higher than those of Col-0, indicating overexpression of ERF1 increased seedlings’ sensitivity to ethylene (Figure 5D).

To further investigate the genetic link of MYC2, MYC3, and ERF1, *erf1-3* and *ERF1-OE1* were hybridized with null mutants of MYC2 or MYC3, and materials with different genetic backgrounds were identified from their offsprings. The response of etiolated seedlings to ACC showed that the sensitivity of three double-null mutants (*myc2-1erf1-3*, *myc2-2erf1-3*, and *myc3erf1-3*) was similar to that of *erf1-3*. However, seedlings of *myc2-1ERF1-OE1, myc2-2ERF1-OE1*, and *myc3ERF1-OE1* showed sensitivity to ACC similar to that of *ERF1-OE1* (Figure 6). These results further demonstrate that MYC2 and MYC3 act upstream of ERF1.

### 2.5. MYC2 and MYC3 Are Involved in Ethylene-Regulated Gene Expression

To better understand the role of MYC2 and MYC3 in ethylene-regulated hypocotyl growth, high-throughput RNA sequencing was used to analyze changes in gene expression pattern after ACC treatment in both Col-0 and *myc2myc3*. Following 25 μM ACC treatment, 1651 differentially expressed genes (DEGs) (fold change > 2) were identified in Col-0 seedlings, with 853 genes up-regulated and 798 genes down-regulated. In *myc2myc3* seedlings, 1559 DEGs were identified, with 636 genes up-regulated and 923 genes down-regulated (Figure 7A). Of these genes, 724 were differentially expressed both in Col-0 and *myc2myc3* seedlings (Figure 7B). Gene ontology (GO) analysis revealed that in Col-0 seedlings, DEGs were enriched in response to stress, plant hormone signaling, cell wall organization, and cellular metabolism, while in *myc2myc3* seedlings, DEGs were enriched in response to ethylene, water or salt stress, plant hormone signaling, and secondary metabolism (Figure 7C and Appendix A).

To further verify the role of MYC2 and MYC3 in ethylene-regulated gene expression, DEGs with a fold change >5 in both Col-0 and *myc2myc3* were analyzed in detail. DEGs that were up-regulated in Col-0 while not up-regulated in *myc2myc3* are involved in cell wall organization, oxidation and reduction, stress response, gene expression regulation, etc. (Table 1). This suggests that MYC2 and MYC3 may have a positive role in the ethylene-regulated expression of these genes. DEGs that were down-regulated in Col-0 while not down-regulated in *myc2myc3* are involved in cellular metabolism, growth and development, signal transduction, oxidation and reduction, gene expression regulation, etc. (Table 2). This suggests that MYC2 and MYC3 may play a negative role in the ethylene-regulated expression of these genes. Further analysis revealed that DEGs that were up-regulated in *myc2myc3* while not up-regulated in Col-0 are involved in cell wall organization, cellular metabolism, signal transduction, gene expression regulation, etc. (Appendix A). DEGs that were down-regulated in *myc2myc3* while not down-regulated in Col-0 are involved in cellular metabolism, hormone signaling, oxidation and reduction, stress response, detoxification, etc. (Appendix A).

## 3. Discussion

MYCs are important transcription factors involved in JA, ABA, and GA signaling as well as photoreceptor-mediated light signaling [43,44,45]. In 2020, Yi et al. reported that light induces JA synthesis, and the binding of JA to its receptor COI1 activates MYC2, MYC3, and MYC4. These transcription factors bind to the promoter of *HY5*, leading to suppressed hypocotyl elongation [23]. The role of MYCs in skotomorphogenesis has been rarely reported. We investigated the growth of etiolated seedlings of MYCs-null mutants (including *myc2*, *myc3*, *myc4*, and *myc5*) and found that the etiolated hypocotyls of *myc2* and *myc3* were significantly longer than those of the wild type, indicating their novel role in regulating etiolated hypocotyl elongation.

As a typical ethylene-induced response, the “triple response” had been used to verify the involvement of ethylene in physiological processes. As a precursor of ethylene synthesis, ACC has been used in studies about the physiological function and mechanism of ethylene [11,46,47]. In this work, etiolated hypocotyls of *myc2* or *myc3* seedlings showed increased sensitivity to ACC, illustrating that MYC2 and MYC3 play negative roles in ethylene-inhibited etiolated hypocotyl elongation. In un-stimulated seedlings, endogenous growth substrates including ethylene finely modulate seedling growth, ensuring the steady growth of seedlings in darkness [3,4,48]. MYC2 and MYC3 may be downstream targets of ethylene signaling, acting as growth suppressors and playing synergistic roles.

The signal transduction process of ethylene-induced triple response has been illustrated. Ethylene binds its receptors, releases their inhibitory effect on CTR1 and EIN2, and then regulates cell elongation through transcription factors EIN3/EIL1 and ERFs, which ultimately inhibits hypocotyl elongation [6]. The results here showed that ethylene inhibited cell elongation rather than proliferation since cell number was not altered after ACC treatment, consistent with the reported data [49]. Following the illustration of the framework of ethylene signal transduction, more detailed mechanisms were revealed; e.g., EIN3 induced the transcription of *PIF3*, and PIF3 regulated expression of a microtubule-related protein, MDP60; led microtubule re-organization; and enhanced hypocotyl growth [50]. Cross-talk between ethylene-CTR1 and the Glc-TOR signaling pathway regulated the etiolated hypocotyl elongation by affecting multiple phosphorylation sites in EIN2 [51]. Moreover, ethylene regulates hypocotyl elongation by crosstalk with light signaling components, e.g., PhyB, PIF3, COP1, and HY5. Ethylene stimulates PIF3 to promote hypocotyl elongation under light, whereas photoactivated phyB may attenuate the promoting effect of ethylene. In darkness, ethylene inhibited hypocotyl elongation by stimulating ERF1 and WDL5. When seedlings grow up to the soil surface, increased light inhibits ethylene synthesis and eliminates the inhibitory effect of EIN3 on hypocotyl elongation [6]. Dozens of transcription factors are involved in ethylene signal transduction, e.g., PIFs, MYCs, HSFs, and CBFs, some of which are bHLH family members (PIF1/3/4/5, MYC2/3/4, RLS2/4, RHD6, etc.) [52]. Nevertheless, so far, the position and role of MYC2 or MYC3 in the ethylene signaling pathway have not been elucidated. The results here are of value for elucidating the role of the two transcription factors in ethylene signaling as well as their role in the regulation of growth and development.

To verify the role of the two MYCs in ethylene signaling, the relationship of MYC2 or MYC3 with ERF1 and EIN3 was intensively investigated. The interaction between EIN3 and MYC2, MYC3, and MYC4 have been reported; EIN3 also binds and regulates the transcription of ERF1 [15,53]. However, the relationship between MYC2 or MYC3 and ERF1 has not been revealed so far. As a crucial component in ethylene signaling, EIN3 can be stimulated by EIN2 and then stimulates multiple downstream signaling agents, including the apical curvature regulator *HLS1* [38]; the root hair elongation factor *RSL4* [54]; the senescence-related genes *ORE1*, *SAG29* [55], and *ESE1* [56]; cold stress-related genes *CBF1/2/3* [57]; and pathogenicity-related genes ERF1 and *ORA59* [42], which in turn regulate growth of vegetative organs (e.g., apical hook, root hair, and leaves) as well as response to (a)biotic stresses [58]. The results in this work suggest that MYC2 and MYC3 are closely associated with EIN3 and involved in ethylene-regulated hypocotyl growth, providing new evidence to clarify the mechanism of EIN3-involved regulation of plant growth.

AP2/ERF are essential transcription factors, which are involved in multiple physiological functions’ regulation [59,60,61]. ERF1 is involved in plant growth, development, and stress responses, which are modulated by various phytohormones, e.g., ethylene, JA, auxin, and ABA (abscisic acid) [42,62,63,64,65]. Expression of ERF1 is regulated by EIN3/EIL1, ABI3, or NIGT1.4 at the transcriptional level [65,66,67]. After translation, UBC18 or MPK3/MPK6 may regulate ERF1 activity by protein modification, e.g., ubiquitination and phosphorylation. The abundance and activity of ERF1 directly affect cell metabolism, proliferation, and differentiation [68,69]. The regulatory mechanism of ERF1 expression has also been intensively explored [64,65,70]. In this work, ERF1 expression was significantly increased in *MYC2-* and MYC3-null mutants, indicating that MYC2 and MYC3 may be involved in regulating ERF1 expression. Subsequently, EMSA and ChIP assays showed that MYC2 and MYC3 directly bind the promoter of ERF1 and inhibit its expression. The data indicated that MYCs may act at upstream of ERF1, providing novel clues for understanding the regulation mechanism of ERF1 expression. An assay of the relationship among EIN3, MYCs, and ERF1 showed that EIN3 suppressed the effect of MYC2 or MYC3 on ERF1 expression. MYC2 and MYC3 were revealed to antagonize the effect of EIN3 on the expression of *HLS1* [41]. Here, it was verified that EIN3 inhibits the binding of MYC2 or MYC3 with ERF1 promoter, clarifying the inhibitory mechanism of EIN3 on MYCs-regulated ERF1 expression. The more detailed interacting mechanism needs to be clarified in the coming days.

The analysis of DEGs after ACC treatment indicated that MYC2 and MYC3 may be essential components in ethylene-regulated gene expression. The complicated data showed that MYC2 and MYC3 may play positive roles in some genes’ expression while playing negative roles in other genes’ expression. The data suggest that MYC2 and MYC3 affect the expression of multiple sets of functional genes that are involved in cellular metabolism, cell wall organization, growth and development, and stress responses. These effects may ultimately impact the elongation of hypocotyls. In the double-null mutant, impaired regulation of gene expression by MYC2 and MYC3 resulted in altered gene expression pattern and the enhanced sensitivity to ethylene. The genes involved in the biosynthesis of glucosinolates were found to be down-regulated in *myc2myc3*, which is consistent with previous studies showing that MYCs play a role in promoting the expression of genes related to glucosinolate production [71,72]. As precursors of multiple signal agents, including JA, SA (salicylic acid), and flavonoids, phenylpropanoids play essential roles in plants’ resistance to multiple stresses. Several MYCs have been reported to be involved in regulating the biosynthesis of phenylpropanoids [73]. The results here further revealed the involvement of MYC2 and MYC3 in phenylpropanoids biosynthesis. These findings offer new insights into how MYCs are involved in resisting both biotic and abiotic stresses, although the mechanism needs to be intensively investigated in the future.

The data in the work suggest that EIN3 may be located upstream of the two MYCs and inhibit their transcription activity. MYC2 and MYC3 inhibit ERF1 expression. When MYC2 and MYC3 are mutated, ERF1 gene expression may be enhanced, as the hypocotyls’ ethylene responsiveness is markedly increased (Figure 8).

## 4. Materials and Methods

### 4.1. Plant Materials

All Arabidopsis materials were derived from the ecotype Col-0 background. The mutants, including *myc2-1* (SALK_017005), *myc2-2* (SALK_061267), and *myc3* (SALK_01 2763), were obtained from the Arabidopsis Biological Resource Center (Columbus, OH, USA). The double-null mutant *myc2myc3* was obtained by crossing *myc2-1* and *myc3*. The ERF1-knock-out mutants, *erf1-1* and erf1-2, were obtained by CRISPR/Cas9 mutation in Col-0. Two more ERF1-knock-out mutants, *erf1-3* and *erf1-4*, were obtained by *RNAi.*

Seeds were surface-sterilized with 70% ethanol after being washed with sterilized in water three times; then, seeds were sown on a solid medium consisting of 1/2 MS salt, 1% (*w*/*v*) agar, and 3% (*w*/*v*) sucrose and then cultured under light (22 °C, 16/8 h light/dark, 130 μmol∙m^−2^∙s^−1^). For etiolated hypocotyl growth measurement, culture dishes were stored at 22 °C under light for 6 h and then coated with foil and cultured at 22 °C for 4 days. 

After being cultured for some time, seedlings were photographed using a scanner (Perfection V39II, Epson, Nagano, Japan), and images were analyzed by using Image J (V1.8.0) software to measure hypocotyl length. Statistical analysis of the data was performed using IBM SPSS Statistics 22.0 software (IBM, New York, NY, USA).

### 4.2. Gene Transformation

The *pSuper::MYC2* and *pSuper::MYC3* plasmids were introduced into *Agrobacterium tumefaciens* (strain GV3101) using the freeze–thaw method. The fused genes were then transformed into wild-type (Col-0) plants through floral dipping. Transgenic plants were screened using 1/2 MS medium containing 100 mg∙mL^−1^ kanamycin.

### 4.3. PCR Analysis

Total RNA was isolated from seedlings for quantitative real-time PCR analyses, which were performed using CFX96^TM^ real-time PCR system (Bio-Rad, Hercules, CA, USA). The primers used for subsequent detection of ERF1 expression were 5’-ACACGTGCGGGATATCAAATA-3’ and 5’-ACACGTGCGGGATATCAAATA-3’. *ACTIN2* was used as a reference (5’-GGTAACATTGTGCTCAGTGGTGG-3’ and 5’-AACGACCTTAATCTTCATGCTGC-3’). Three biological replicates and three technical replicates were used each time. Mean ± SE was calculated and statistically analyzed.

### 4.4. Electrophoretic Mobility Shift Assay (EMSA)

EMSA was performed according to the reported method [49]. The recombinant GST-MYC2 and GST-MYC3 protein was expressed in *E. coli* and then extracted and purified. EMSA was conducted using the Light Shift Chemiluminescent EMSA Kit (Thermo Scientific, Rockford, IL, USA) with biotin-labeled and cold probes.

Nucleotide sequences of the double-stranded oligonucleotides were as follows: ERF1 P1: 5’-AAAACTTTGAACACGTGCGGGATATCAA-3’ and 5’-TTGATATCCCGCACGTGTTCAAAGTTTT-3’. ERF1 P2: 5’-GAAAAAATGGCACATGAAGTATCTTT-3’ and 5’-AAAGATACTTCATGTGCCATTTTTTC-3’. ERF1 P3: 5’-ACTCAGGATGCATGTGATGATGTGTA-3’ and 5’-TACACATCATCACATGCATCCTGAGT-3’. ERF1 P4: 5’-TTATCTTCTAAACATGAGATGGCTCG-3’ and 5’-CGAGCCATCTCATGTTTAGAAGATAA-3’. ERF1 P5: 5’-TTAGTTGCGTATCCGTTCGAATAATT-3’ and 5’-AATTATTCGAACGGATACGCAACTAA-3’. Mutated ERF1 P1: 5’-AAAACTTTGAATCTACACGGGATATCAA-3’ and 5’-TTGATATCCCGTGTAGATTCAAAGTTTT-3’.

### 4.5. Chromatin Immunoprecipitation (ChIP)

The 10-day-old seedlings of *pSuper::MYC2-GFP* or *pSuper::MYC3-GFP* transformant were fixed in 1% formaldehyde and then were homogenized and sonicated. The sheared chromatins were immunoprecipitated, washed, reverse cross-linked, and amplified. The monoclonal anti-GFP antibody was used for immunoprecipitation. Approximately 10% of sonicated but non-immunoprecipitated chromatin was reverse cross-linked and used as an input DNA control. The immunoprecipitated DNA and input DNA were analyzed by using real-time q-PCR. The primers used to detect the MYC2 and MYC3 target ERF1 promoter were 5’-ACACGTGCGGGATATCAAATA-3’ and 5’-CATGCATCCTGAGTCC AGTAG-3’, with *ACTIN2* as a control (5’-GGTAACATTGTGCTCAGTGGTGG-3’ and 5’-AACGACCTTAATCTTCATGCTGC-3’).

### 4.6. Gene Transient Expression in N. benthamiana

*ERF1_pro_::GUS* or *ERF1_mpro_::GUS* with pCAMBIA1391, *pSuper::MYC2*, *pSuper::MYC3,* or *pSuper::EIN3* with pCAMBIA1300 constructs were transformed into *A*. *tumefaciens* (strain GV3101). Cultured *A. tumefaciens* cells were harvested by centrifugation and suspended and diluted with a solution consisting of 10 mM MES, 10 mM MgCl_2_, and 200 mM acetosyringone (pH 5.6) to an optical density (600 nm, OD = 0.7). Cells were injected into *N. benthamiana* leaves by using a needle-free syringe. The GUS activity of the infiltrated leaves was quantitatively assessed.

### 4.7. Yeast Two-Hybrid Assays

The intracellular domain of MYC2, MYC3, or EIN2-C (1255-3886 bp) was cloned into pGBKT7 and pGADT7 vectors. Constructs including MYC2-pGBKT7, MYC3-pGBKT7, and EIN2-C-pGADT7 were co-transformed into yeast strain AH109 and then grown on SD/LW medium (synthetic dextrose medium lacking Leu and Trp). Subsequently, the yeast cells were screened on SD/LWH medium (synthetic dextrose medium lacking Leu, Trp, and His) or SD/LWHA medium (synthetic dextrose medium lacking Leu, Trp, His, and Ade).

## Figures and Tables

**Figure 1 ijms-25-08022-f001:**
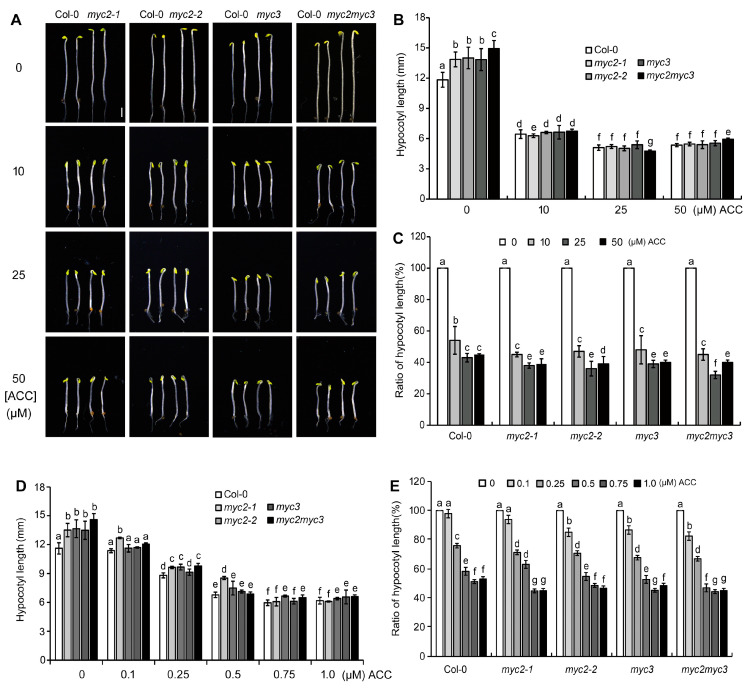
MYC2 and MYC3 negatively regulate ethylene-inhibited elongation of etiolated hypocotyls. (**A**). Photograph of etiolated seedlings grown on 1/2 MS medium containing serial concentration of ACC for 4 days. Scale bar = 1 mm. (**B**,**D**). Mean ± SE hypocotyl length after ACC treatment: (**B**) (10 to 50 μM) and (**D**) (0.1 to 1.0 μM). (**C**,**E**) The ratio ((ACC treatment/control)*100%) of hypocotyl length after ACC treatment: (**C**) (10 to 50 μM) and (**E**) (0.1 to 1.0 μM). In each experiment, at least 40 seedlings were measured. Data from 3 replicates were calculated and statistically analyzed using one-way ANOVA. Different lowercase letters within any data are significantly different based on the Duncan multiple range test at *p* < 0.05.

**Figure 2 ijms-25-08022-f002:**
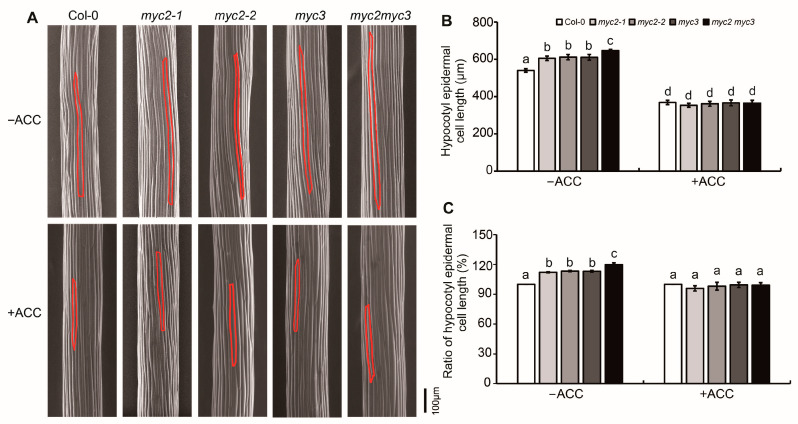
MYC2 and MYC3 negatively regulated ethylene-inhibited elongation of etiolated hypocotyl epidermal cells. (**A**). SEM photograph of epidermal cells of etiolated seedlings grown on MS medium containing 25 μM ACC for 4 days. In each photograph, the red line encircles a single epidermal cell. (**B**,**C**) Mean ± SE of hypocotyl epidermal cell length (**B**) and the ratio of hypocotyl epidermal cell length after/before ACC treatment (**C**). In each experiment, at least 10 seedlings were investigated. Data from 3 replicates were calculated and statistically analyzed using one-way ANOVA. Different lowercase letters within any data are significantly different based on the Duncan multiple range test at *p* < 0.05.

**Figure 3 ijms-25-08022-f003:**
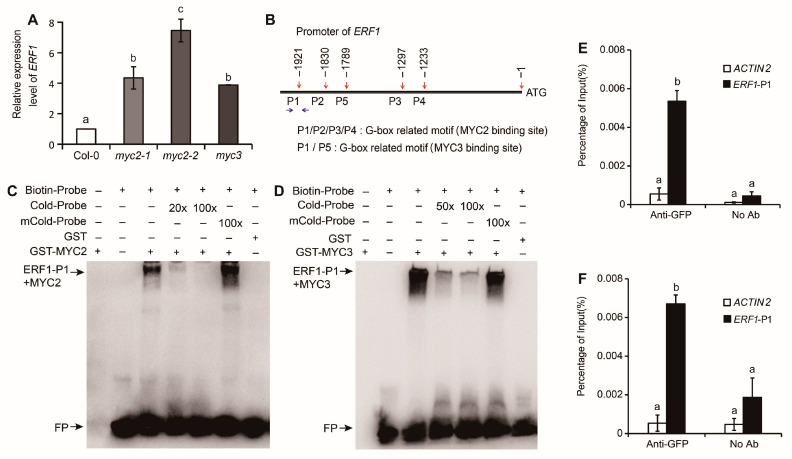
ERF1 is a target of MYC2 and MYC3. (**A**) ERF1 expression level in etiolated wild-type and mutant seedlings. (**B**) Five predicted MYCs-binding sites (marked by red arrows) in the promoter region of ERF1. The region amplified by ChIP-qPCR assay (ERF1-P1) is marked by blue arrows. (**C**,**D**) EMSA results of the binding of MYC2 (**C**) or MYC3 (**D**) with ERF1 promoter. (**E**,**F**) ChIP-qPCR assay of MYC2 and MYC3 binding to ERF1-P1 in vivo. In figure (**A**,**E**,**F**), the mean ± SE of the data from three replicates was calculated and statistically analyzed using one-way ANOVA. Different lowercase letters within any data are significantly different based on the Duncan multiple range test at *p* < 0.05.

**Figure 4 ijms-25-08022-f004:**
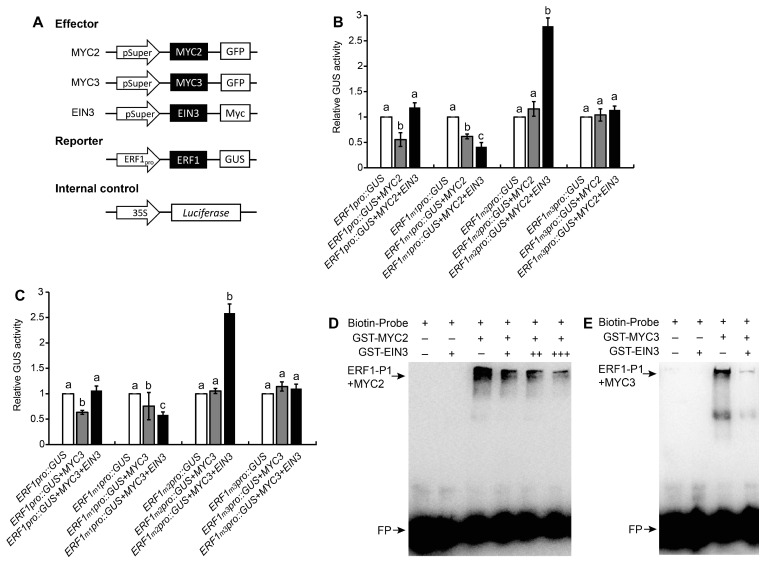
EIN3 suppressed the transcriptional activity of MYC2 or MYC3. (**A**) Schematic image of the reporter and effector used in transient transactivation assays. (**B**,**C**) Transient expression of MYC2, MYC3, *EIN3*, and *ERF1_pro_::GUS* or *ERF1_mpro_::GUS* (*ERF1_m1pro_*: EIN3 binding sites were mutated; *ERF1_m2pro_*: P1 was mutated; *ERF1_m3pro_*: EIN3 binding sites and P1 were both mutated) in *N. benthamiana* leaves. *ERF1_pro_::GUS* or *ERF1_mpro_::GUS* expression level was set to 1. Mean ± SE of data from three replicates was calculated and statistically analyzed using one-way ANOVA. Different lowercase letters within any data are significantly different based on the Duncan multiple range test at *p* < 0.05. (**D**,**E**) EIN3 blocked the binding of MYC2 or MYC3 with ERF1 promoter. The binding of MYC2 (**D**) or MYC3 (**E**) with P1 was detected using EMSA.

**Figure 5 ijms-25-08022-f005:**
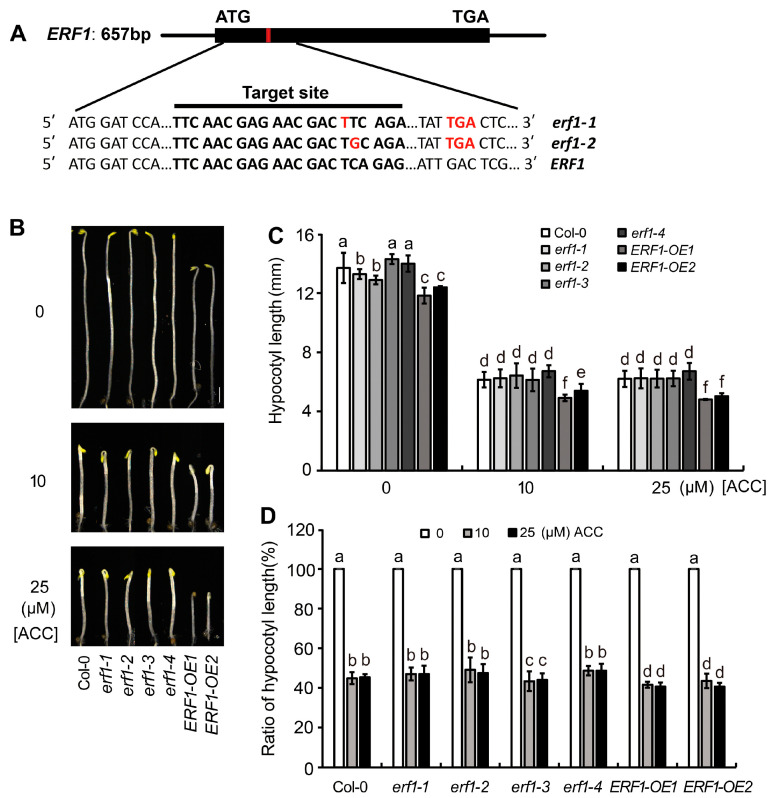
ERF1 is involved in ethylene-regulated elongation of etiolated hypocotyl. (**A**). Schematic diagram of ERF1 gene structure. The black boxes note the exons. Red frame notes the point of a T-base insertion. Sequence of gRNA target site of ERF1 is noted below. (**B**) Overexpression of ERF1 suppressed elongation of etiolated hypocotyls. Seedlings were grown on 1/2 MS medium in darkness for 4 days. (**C**,**D**). The effect of ACC treatment on elongation of etiolated hypocotyl of ERF1-null mutants or OE lines. Hypocotyl length (**C**) and ratio of hypocotyl length (**D**) are shown, respectively. In each experiment, at least 40 seedlings were measured. Data from 3 replicates were calculated and statistically analyzed using one-way ANOVA. Different lowercase letters within any data are significantly different based on the Duncan multiple range test at *p* < 0.05.

**Figure 6 ijms-25-08022-f006:**
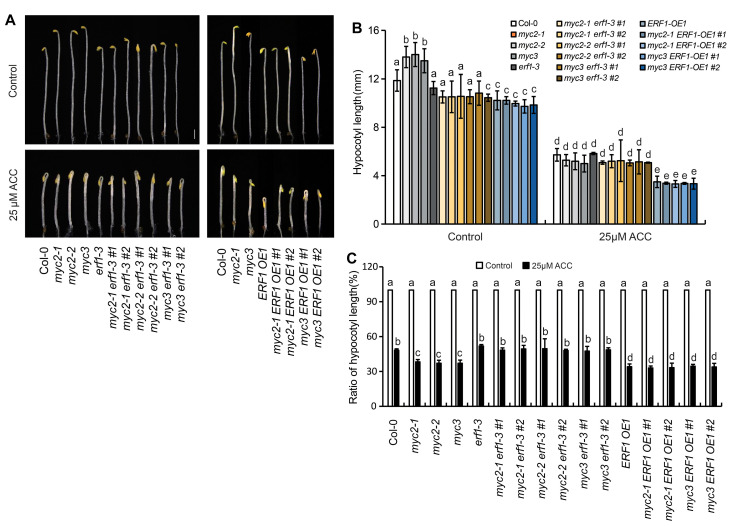
MYC2 and MYC3 function at upstream of ERF1. Null mutants of MYC2 or MYC3 were hybridized with *erf1-3* or *ERF1-OE1*. Effect of ACC on seedlings of hybrid progenies was detected. (**A**) Photograph of etiolated seedlings grown on ACC-containing medium for 4 days. In figure (**B**,**C**), hypocotyl length (**B**) and ratio of hypocotyl length (**C**) after/before ACC treatment are shown, respectively. In each experiment, at least 40 seedlings were measured. Data from 3 replicates were calculated and statistically analyzed using one-way ANOVA. Different lowercase letters within any data are significantly different based on the Duncan multiple range test at *p* < 0.05.

**Figure 7 ijms-25-08022-f007:**
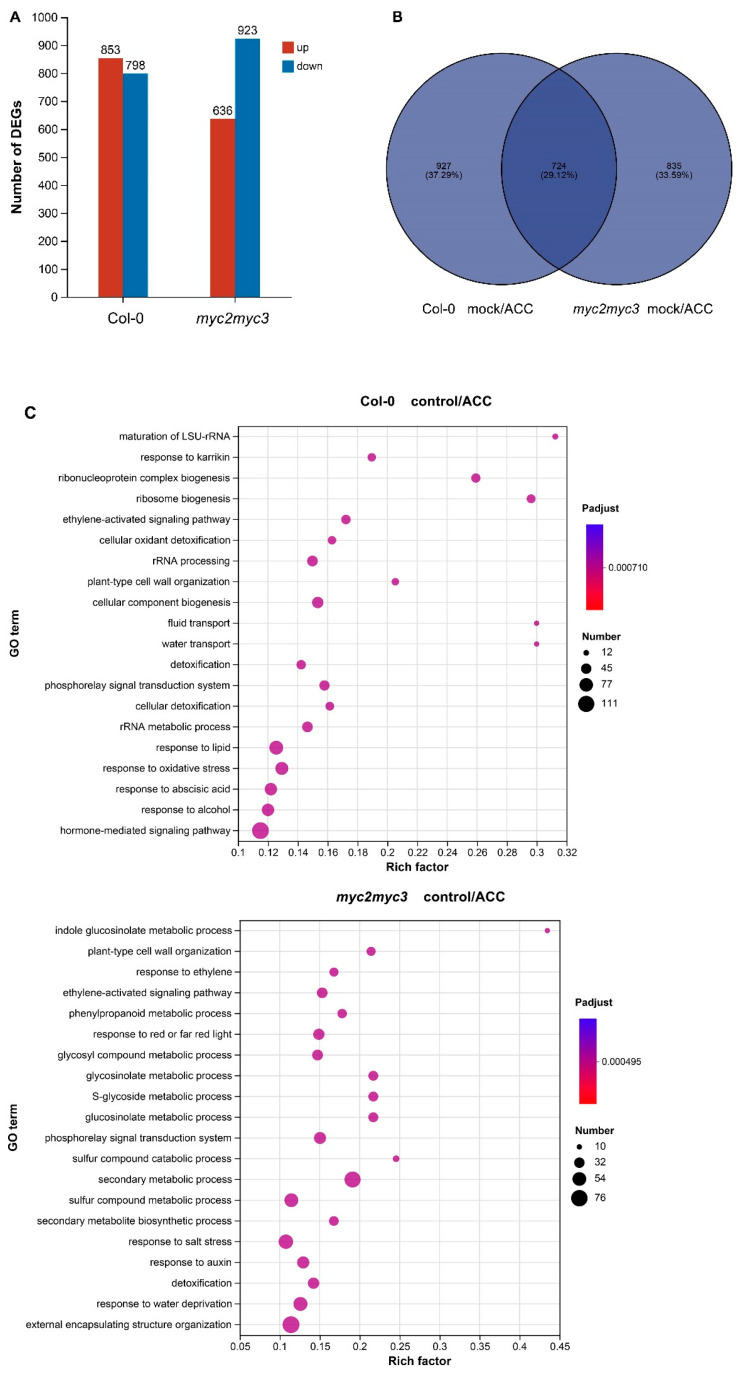
MYC2 and MYC3 are involved in ethylene-regulated expression of functional genes. Etiolated seedlings were grown on a medium containing 25 μM ACC for 4 days. High-throughput RNA-seq was used to analyze the expression pattern of functional genes. (**A**) Number of ethylene-regulated DEGs in Col-0 and *myc2myc3*. (**B**) Venn analysis of DEGs. (**C**) GO (gene ontology) analysis of DEGs.

**Figure 8 ijms-25-08022-f008:**
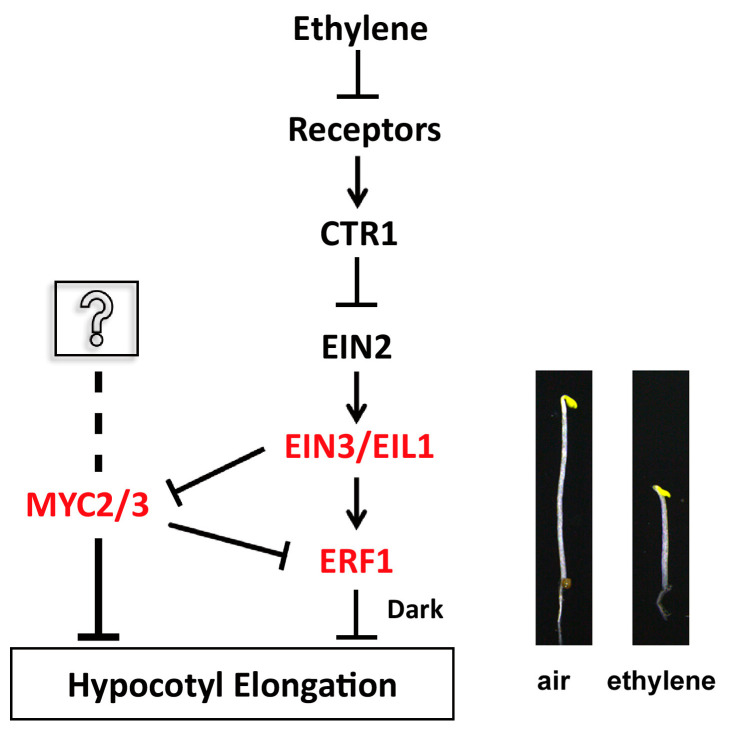
Model of the function of MYC2 or MYC3 in ethylene-inhibited elongation of etiolated hypocotyls. Arrows and bars indicate positive and negative regulation, respectively.

**Table 1 ijms-25-08022-t001:** DEGs that were up-regulated in Col-0 while not up-regulated in *myc2myc3* after ACC treatment.

Functional Category	Gene_id	Gene Name	Gene Description
Cell wall organization	AT1G54970	PRP1	Involved in cell wall structure formation, inducible by ethylene or auxin
	AT1G26250		Proline-rich extensin-like protein
	AT1G02460		Pectin lyase-like superfamily protein
	AT1G62980	EXPA18	Expansin-A18, involved in cell wall loosening
Growth and development	AT5G49870	JAL48	Involved in plant growth and development
	AT1G18630	RBG6	Involved in regulation of growth and development
Hormone metabolism or signaling	AT1G04180	YUC9	Involved in auxin synthesis
	AT4G00880		SAUR-like auxin-responsive protein
	AT5G25350	EBF2	EIN3-binding F-box protein 2, involved in ethylene signaling
Oxidation reduction	AT3G49960	PER35	Peroxidase 35
	AT1G69880	TRX8	Thioredoxin H8, involved in cell redox homeostasis
	AT5G25180	CYP71B14	Cytochrome P450 family protein
	AT4G20240	CYP71A27	Cytochrome P450 family protein
	AT4G26010		A peroxidase
Signal transduction	AT4G04220	AtRLP46	Receptor-like protein 46
	AT3G23120	AtRLP38	Receptor-like protein 38
	AT1G03010		An NPH3 family protein, involved in response to light stimulus
Stress response	AT2G05230		DNA-J heat-shock protein
	AT1G53130	GRI	Involved in extracellular ROS-induced cell death
	AT4G09435		Involved in disease resistance
	AT4G38410		Dehydrin family protein, involved in response to water stress
	AT5G14200	IMDH3	3-isopropylmalate dehydrogenase, involved in glucosinolate synthesis
Transcription factor	AT5G57520	ZFP2	Zinc finger protein 2
	AT1G03790	SOM	Zinc finger CCCH domain-containing protein 2
	AT3G62090	PIF6	Myc-related bHLH transcription factor
	AT5G17430	BBM	AP2-domain containing protein
	AT3G60490	ERF035	Ethylene-responsive transcription factor 035

**Table 2 ijms-25-08022-t002:** DEGs which were down-regulated in Col-0 while not down-regulated in *myc2myc3* after ACC treatment.

Functional Category	Gene_id	Gene Name	Gene Description
Cell wall organization	AT3G50220	IRX15	Involved in xylan synthesis and deposition
	AT3G55090	ABCG16	Required for synthesis of cell wall
Cellular metabolism	AT1G05675	UGT74E1	UDP-Glycosyltransferase superfamily protein
	AT4G25835		AAA-ATPase, P-loop containing NTP hydrolases
	AT4G25150		Acid phosphatase-like protein
	AT1G06520	GPAT1	Glycerol-3-phosphate acyltransferase 1
	AT1G66040	ORTH4	E3 ubiquitin-protein ligase
	AT1G06030		Fructokinase-2
	AT5G05270	CHI3	Chalcone–flavonone isomerase 3, involved in response to karrikin
	AT1G65060	4CL3	4-coumarate-CoA ligase 3, involved in phenylpropanoid synthesis
	AT1G72520	LOX4	Lipoxygenase 4, involved in growth and defense response
	AT2G28210	ATACA2	Alpha carbonic anhydrase 2
Growth and development	AT2G29950	EFL1	ELF4-LIKE 1, involved in circadian and flowering
	AT3G18217	MIR157C	MIR157C, involved in regulating vegetative phase
	AT5G51720	NEET	Involved in plant development
	AT1G43790	TED6	Involved in tracheary element differentiation
	AT2G46830	CCA1	Involved in regulating circadian rhythms
	AT1G65620	AS2	Involved in formation of a symmetric flat leaf lamina
	AT1G52690	LEA7	Late embryogenesis abundant protein 7
Hormone metabolism or signaling	AT2G14960	GH3.1	IAA-amido synthetase
	AT2G16660		Involved in response to karrikin
	AT5G66260		SAUR-like auxin-responsive protein
Oxidation reduction	AT3G49110	PER33	Peroxidase 33
	AT1G32780		Alcohol dehydrogenase-like 3
	AT3G44970		Cytochrome P450 family protein
	AT2G30540	GRXS9	Monothiol glutaredoxin-S9
	AT5G25130	CYP71B12	Cytochrome P450 family protein
Signal transduction	AT3G15050	IQD10	A calmodulin binding protein
	AT5G15430		A calmodulin-binding protein-like protein
	AT3G01830	CML40	Calmodulin-like protein 40
	AT1G21550	CML44	Calmodulin-like protein 44
	AT2G24850	TAT3	A tyrosine aminotransferase that is responsive to jasmonic acid
	AT5G43300	GDPD3	PLC-like phosphodiesterase
	AT3G45430	LECRK15	L-type lectin-domain containing receptor kinase I.5
	AT3G45390	LECRK12	L-type lectin-domain containing receptor kinase I.2
	AT5G55830	LECRKS7	L-type lectin-domain containing receptor kinase S.7
Stress response	AT1G12280	SUMM2	A disease resistance protein, involved in defense response
	AT3G59930		Defensin-like protein 206
	AT2G03020		Heat-shock protein HSP20
Transcription factor	AT5G61620		myb-like transcription factor family protein
	AT1G73870	COL7	Zinc finger protein CONSTANS-LIKE 7
	AT1G62700	NAC026	NAC (No apical meristem) domain transcriptional regulator
	AT5G64750	ABR1	Ethylene-responsive transcription factor
	AT1G22810	ERF019	Ethylene-responsive transcription factor

## Data Availability

Data available on request from the authors.

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
