# Peer review of "Arabidopsis thaliana MYC2 and MYC3 Are Involved in Ethylene-Regulated Hypocotyl Growth as Negative Regulators"

_ijms, 2024, doi:10.3390/ijms25158022_

Round 1

Reviewer 1 Report

Comments and Suggestions for Authors

The study investigates the role of MYC transcription factors, specifically MYC2 and MYC3, in the ethylene-mediated regulation of hypocotyl elongation in Arabidopsis thaliana. Ethylene, a plant hormone, is known to inhibit hypocotyl elongation in dark-grown seedlings. The findings demonstrate that MYC2 and MYC3 act as negative regulators of ethylene-suppressed hypocotyl elongation. Loss-of-function mutants of MYC2 or MYC3 exhibited longer hypocotyls compared to wild-type seedlings when exposed to ACC (1-aminocyclopropane-1-carboxylate), an ethylene precursor. Moreover, single- or double-null mutants of MYC2 and MYC3 showed an enhanced response to ACC, indicating a more pronounced hypocotyl elongation in the absence of these transcription factors.

Further investigation revealed that MYC2 and MYC3 directly bind to the promoter region of ERF1, a key gene involved in ethylene signaling, thereby suppressing its expression. Additionally, EIN3, another crucial component in ethylene signaling, interacts with MYC2 or MYC3, inhibiting their binding to ERF1's promoter. This interaction between EIN3 and MYC transcription factors highlights a regulatory mechanism where EIN3 modulates the activity of MYC2 and MYC3 in ethylene signaling pathways. Overall, the study identifies a novel role for MYC2 and MYC3 in the ethylene-regulated expression of functional genes, particularly in the context of hypocotyl elongation. These findings significantly contribute to the understanding of ethylene signal transduction mechanisms in plants, emphasizing the intricate regulatory networks involving transcription factors like MYCs and their interactions with key players such as EIN3.

Some comments:

·         Authors need to provide significance and explain implications of the study in abstract + conclusions.

·         Provide statistical information in abstract and discuss shortcomings of the study.

Comments on the Quality of English Language

English can be improved.

Author Response

Q1. Authors need to provide significance and explain implications of the study in abstract + conclusions.

Reply: Thanks for your warm advice. We have provided some comments in the abstract and the discussion. 

Q2. Provide statistical information in abstract and discuss shortcomings of the study.

Reply: Thanks for your warm advice. We have provided statistical information in the abstract and discussed the shortcomings in the discussion section. See detail in line 411-412, line 430-431. 

Reviewer 2 Report

Comments and Suggestions for Authors

The manuscript describes the roles as negative regulators of MYC2 and MYC3 transcription factors in Arabidopsis hypocotyl growth via ethylene signaling. MYC2 and MYC3 are central transcription factors of jasmonic acid signaling. Many papers have reported the antagonistic regulation of jasmonic acid and ethylene. However, this manuscript provides novel insight into the fact that MYC2 and MYC3 directly suppress the gene expression of ERF1, which encodes the ethylene-responsive transcription factor. The manuscript shows solid genetic evidence of the roles as negative regulators of MYC2 and MYC3. However, there is a lack of explanation, experimental data, and discussion regarding how MYC2 and MYC3 bind to and suppress ERF1 expression. There are several concerns should be addressed as listed below.

1)     (Figures 1 and S1) The low-concentration ACC treatment data shown in Figure S1 more clearly shows ethylene hypersensitivity in myc mutants. The effect of ACC is saturated in the high-concentration conditions shown in Figure 1. Please consider swapping Figure 1 and Figure S1.

2)     Can the author explain the different responses of myc2-1 and myc2-2 mutants to low concentrations of ACC treatment (Figure S1)? How are the mutations of these two mutants different?

3)     The description of the ERF1 promoter is missing as listed below:

- The specific sequence for the candidates of the MYC binding sequence described in lines 185-188 should be described.

-The specific sequence changes for the G-box mutations introduced in the EMSA and reporter assays should be described in the Materials and Methods and figure legends.

- The EIN3 binding sequence reported by Solano et al. (cited as 15) is -1213 to -1179 of the ERF1 promoter. Please clarify the positional relationship with the MYC binding sequence shown in Figure 3B.

4)     Is the MYC-EIN3 fusion gene used in Figure 4A? Why not express MYC and EIN3 independently as effectors? If it is a fusion protein, there is a possibility that the original function of the interaction will not be shown, but the artificial effect will be shown.

5)     Figure 4D shows that EIN3 suppresses MYC bindings to the P1 region of the ERF1 promoter without EIN3 binding, suggesting that EIN3-MYC interaction inhibits DNA binding activities of MYCs. Using the ein3 mutant strain as a background, the ChIP in Figure 3E should show that EIN3 inhibits MYC DNA binding in vivo.

6)     The regions amplified by ChIP-PCR (Figure 3E, F) should be shown in Figure 3B.

Author Response

Q1. (Figures 1 and S1) The low-concentration ACC treatment data shown in Figure S1 more clearly shows ethylene hypersensitivity in myc mutants. The effect of ACC is saturated in the high-concentration conditions shown in Figure 1. Please consider swapping Figure 1 and Figure S1.

Reply: Thanks for your advice. In the revised manuscript, Figure1 and Figure S1 have been merged. See detail in page 4.

Q2.  Can the author explain the different responses of myc2-1 and myc2-2 mutants to low concentrations of ACC treatment (Figure S1)? How are the mutations of these two mutants different?

Reply: The two mutants, myc2-1 and myc2-2, were ordered from ABRC and both of the two mutants are derived from T-DNA insertion. We observed that the response differed between myc2-1 and myc2-2. Sometimes myc2-1 showed higher sensitivity to ACC than myc2-2, while at other times myc2-2 showed slightly higher sensitivity than myc2-1. We suspect that this discrepancy may be due to experimental data errors. However, both mutants showed enhanced sensitivity to ACC compared to the wild type.

Q3. The description of the ERF1 promoter is missing as listed below:

  • The specific sequence for the candidates of the MYC binding sequence described in lines 185-188 should be described.

Reply: The specific sequences have been showed in the text. See detail in line 188-194.

(2) The specific sequence changes for the G-box mutations introduced in the EMSA and reporter assays should be described in the Materials and Methods and figure legends.

Reply: The mutated sequence has been added into the Material and Methods, section 4.4. See detail in line 483-484.

(3) The EIN3 binding sequence reported by Solano et al. (cited as 15) is -1213 to -1179 of the ERF1 promoter. Please clarify the positional relationship with the MYC binding sequence shown in Figure 3B.

Reply: The EIN3 binding sequence reported by Solano et al. [15] is located between positions -1213 and -1179 in the ERF1 promoter. It is situated between P4 and the initiator and does not overlap with the MYC binding sites.

Q4. Is the MYC-EIN3 fusion gene used in Figure 4A? Why not express MYC and EIN3 independently as effectors? If it is a fusion protein, there is a possibility that the original function of the interaction will not be shown, but the artificial effect will be shown.

Reply: Figure 4A, in the pSuper:: Myc-EIN3 construct, Myc is a tag in the pCAMBIA1300 vector.  In the revised Figure 4A, “Myc” has been deleted. See detail in page 8, figure 4.

Q5. Figure 4D shows that EIN3 suppresses MYC bindings to the P1 region of the ERF1 promoter without EIN3 binding, suggesting that EIN3-MYC interaction inhibits DNA binding activities of MYCs. Using the ein3 mutant strain as a background, the ChIP in Figure 3E should show that EIN3 inhibits MYC DNA binding in vivo.

Reply: Thanks for your helpful advice. Figure 4D just preliminary indicated that EIN3 suppressed the DNA binding activity of MYC2/3. In ongoing work, we are trying to obtain MYC2/MYC3 overexpression lines using the ein3 mutant as a background, which can be used in the following ChIP-qPCR assay. We hope we can put the results into future publication.

Q6.  The regions amplified by ChIP-PCR (Figure 3E, F) should be shown in Figure 3B.

Reply: Thanks for your warm reminder. The region has been marked in the Figure 3B and noted in the figure legend. See detail in page 7.

Reviewer 3 Report

Comments and Suggestions for Authors

Authors investigated the growth of A. thaliana etiolated seedlings of MYCs null mutants (including myc2myc3myc4,myc5) and found that the etiolated hypocotyls of myc2 and myc3 were significantly longer than those of wild type. Also, etiolated hypocotyls of myc2 or myc3 seedlings increased sensitivity to ACC, showing that MYC2 and MYC3 play negative roles in ethylene-inhibited etiolated hypocotyl elongation. These data strongly suggest that MYC2 and MYC3, act as negative regulators in ethylene-suppressed hypocotyl elongation, which is a novel role in ethylene signal transduction. 

Materials and methods of Figure S5, Yeast two hybridization, were not included. Include procedure and describe SD/LW, SD/LWH, SD/LWHA, AD/BD, AD-T/BD-53, AD-T/BD-Lam, AD-EIN2-C/BD, AD/BD-MYC2, AD/BD-MYC3, AD-EIN2-C/BD-MYC2, AD-EIN2-C/BD-MYC3 in Fig. S5

Regarding the analysis of DEGs in myc2myc3   mock/ACC, genes related with glucosinolate products and mainly with phenylpropanoids are enriched and in Fig. 7C and Fig. S6; however, there is not any comment about although relative literature involving MYC2 and MYC3 with these natural products is abundant. It will be desirable to have some comments on this matter to enrich the document and to glimpse some attractive perspectives.

Author Response

Q1. Materials and methods of Figure S5, Yeast two hybridization, were not included. Include procedure and describe SD/LW, SD/LWH, SD/LWHA, AD/BD, AD-T/BD-53, AD-T/BD-Lam, AD-EIN2-C/BD, AD/BD-MYC2, AD/BD-MYC3, AD-EIN2-C/BD-MYC2, AD-EIN2-C/BD-MYC3 in Fig. S5.

Reply: Thanks for your warm notification. Sorry for making such a silly mistake. In the revised manuscript, a description of the yeast two-hybrid assay has been added in section 4.7. See detail in line 504-511.

Q2. Regarding the analysis of DEGs in myc2myc3 mock/ACC, genes related with glucosinolate products and mainly with phenylpropanoids are enriched and in Fig. 7C and Fig. S6; however, there is not any comment about although relative literature involving MYC2 and MYC3 with these natural products is abundant. It will be desirable to have some comments on this matter to enrich the document and to glimpse some attractive perspectives.

Reply: Thanks for your advice. We have provided some comments in the discussion section. See detail in line 420-430.

Round 2

Reviewer 2 Report

Comments and Suggestions for Authors

The reviewer understood that EIN3 was introduced as a Myc-tagged protein in Figure 4. However, it is noted that the data cannot be interpreted due to the lack of appropriate controls for Figure 4. 

There is a complication of whether EIN3 affects the ERF1 promoter through P1 by inhibiting MYC2 activity or the EIN3 binding motif in P4.

In Figure 4, ERF1p:GUS+MYC and ERF1m1p:GUS+MYC+EIN3 are compared. This is not an appropriate comparison. A comparison of ERF1p:GUS+MYC and ERF1p:GUS+MYC+EIN3 should be made to determine the impact of EIN3 on ERF1p. Then, ERF1m1p:GUS+MYC and ERF1m1p:GUS+MYC+EIN3 should be compared to discuss whether the effect of EIN3 on ERF1p is via P4. A comparison of ERF1m2p:GUS+MYC and ERF1m2p:GUS+MYC+EIN3 would also allow us to discuss whether EIN3 acts on ERF1p via MYC2-P1.

In the revised manuscript, it is difficult to conclude from the experiments shown in Figures 4B and C. Also of concern is the statement "indicating that EIN3 inhibits the transcriptional activity of MYC2 or MYC3." (lines 259-260). In the experiments shown in Figures 4B and C, it is impossible to interpret whether MYC and EIN3 work independently or not. EMSA shows that EIN3 inhibits the DNA-binding ability of MYC. Still, the description that "EIN3 inhibits the transcriptional activity of MYC2 or MYC3" is inappropriate.

Author Response

Thank you for your valuable advice. Based on your guidance, we conducted additional experiments. The results revealed that EIN3 promotes the expression of ERF1 through sites other than the P1 site. When the EIN3 binding site was mutated, MYCs inhibited the expression of ERF1. Furthermore, upon adding EIN3, the expression of ERF1 decreased even further. This suggests that MYCs and EIN3 regulate the expression of ERF1 through different sites, and EIN3 hinders MYCs' transcriptional regulation of ERF1. We have revised Figure 4, the figure legend, and the text (see lines 233-247). See detailed changes on page 8.

Round 3

Reviewer 2 Report

Comments and Suggestions for Authors

The revised manuscript includes additional GUS reporter assay experiments, allowing for a deeper discussion on the transcriptional regulation of ERF by EIN3 and MYC2. The reviewer suggests minor revisions to the description of the results in Figure 4. Experiments using the m1 promoter with a mutation in the EIN binding site show no increase in GUS activity by EIN3; experiments using the m2 promoter with a mutation in the MYC binding site show an increase in GUS activity by EIN3. Thus, the reviewer thinks that EIN3 induces ERF promoter independently of MYC in the GUS reporter assay using Nicotiana benthamiana. Therefore, the possibility that EIN3 acts on the ERF promoter independently of MYC should be described in lines 240-242.

Author Response

Comments: The revised manuscript includes additional GUS reporter assay experiments, allowing for a deeper discussion on the transcriptional regulation of ERF by EIN3 and MYC2. The reviewer suggests minor revisions to the description of the results in Figure 4. Experiments using the m1 promoter with a mutation in the EIN binding site show no increase in GUS activity by EIN3; experiments using the m2 promoter with a mutation in the MYC binding site show an increase in GUS activity by EIN3. Thus, the reviewer thinks that EIN3 induces ERF promoter independently of MYC in the GUS reporter assay using Nicotiana benthamiana. Therefore, the possibility that EIN3 acts on the ERF promoter independently of MYC should be described in lines 240-242.

Response:  Thank you for your warm advice. We have added a sentence (lines 241-244) to address the conclusion.